



# Organic aerosol volatility and viscosity in North China Plain: Contrast between summer and winter

Weiqi Xu[1], Chun Chen[1,2], Yanmei Qiu[1,2], Ying Li[1], Zhiqiang Zhang[1,2], Eleni Karnezi[3,a], Spyros N. Pandis[3], Conghui Xie[1,2,b], Zhijie Li[1,2], Jiaxing Sun[1,2], Nan Ma[4], Wanyun Xu[5], Pingqing Fu[2,6], Zifa Wang[1,2], Jiang Zhu[1], Douglas R. Worsnop[7], Nga Lee Ng[8,9,10], and Yele Sun[1,2,11*]

[1]State Key Laboratory of Atmospheric Boundary Layer Physics and Atmospheric Chemistry, Institute of Atmospheric Physics, Chinese Academy of Sciences, Beijing 100029, China
[2]College of Earth and Planetary Sciences, University of Chinese Academy of Sciences, Beijing 100049, China
[3]Department of Chemical Engineering, Carnegie Mellon University, Pittsburgh, PA, USA
[4]Institute for Environmental and Climate Research, Jinan University, Guangzhou 511443, China
[5]State Key Laboratory of Severe Weather & Key Laboratory for Atmospheric Chemistry, Institute of Atmospheric Composition, Chinese Academy of Meteorological Sciences, Beijing, 100081, China
[6]Institute of Surface-Earth System Science, Tianjin University, Tianjin 300072, China
[7]Aerodyne Research Inc., Billerica, Massachusetts 01821, USA
[8]School of Earth and Atmospheric Sciences, Georgia Institute of Technology, Atlanta, GA 30332, USA
[9]School of Chemical and Biomolecular Engineering, Georgia Institute of Technology, Atlanta, GA 30332, USA
[10]School of Civil and Environmental Engineering, Georgia Institute of Technology, Atlanta, GA 30332, USA
[11]Center for Excellence in Regional Atmospheric Environment, Institute of Urban Environment, Chinese Academy of Sciences, Xiamen 361021, China
[a]now at: Earth Sciences Department, Barcelona Supercomputing Center, BSC-CNS, Barcelona 08034, Spain
[b]now at: State Key Joint Laboratory of Environmental Simulation and Pollution Control, College of Environmental Sciences and Engineering, Peking University, Beijing 100871, China

*Correspondence*: Yele Sun (sunyele@mail.iap.ac.cn)

**Abstract.** Volatility and viscosity have substantial impacts on gas-particle partitioning, formation and evolution of aerosol, and hence the predictions of aerosol related air quality and climate effects. Here aerosol volatility and viscosity at a rural site (Gucheng) and an urban site (Beijing) in North China Plain (NCP) in summer and winter were investigated by using a thermodenuder coupled with high resolution aerosol mass spectrometer. The effective saturation concentration ($C^*$) of organic aerosol (OA) in summer was smaller than that in winter (0.55 μg m$^{-3}$ vs. 0.71-0.75 μg m$^{-3}$), indicating that OA in winter in NCP is more volatile due to enhanced primary emissions from coal combustion and biomass burning. The volatility distributions varied largely different among different OA factors. In particular, we found that hydrocarbon-like OA (HOA) contained more non-volatile compounds compared to coal combustion related OA. The more oxidized oxygenated OA (MO-OOA) showed overall lower volatility than less oxidized OOA (LO-OOA) in both summer and winter, yet the volatility of MO-OOA was found to be relative humidity (RH) dependent showing more volatile properties at higher RH. Our results demonstrated the different composition and chemical formation pathways of MO-OOA under different RH levels. The glass transition temperature ($T_g$) and viscosity of OA in summer and winter are estimated using the recently developed parameterization formula. Our results showed that the $T_g$ of OA in summer in Beijing (291.5 K) was higher than that in





winter (289.7-290.0 K), while it varied greatly among different OA factors. The viscosity suggested that OA existed mainly as solid in winter in Beijing, but as semi-solids in Beijing in summer and Gucheng in winter. These results have important implications that kinetically limited gas-particle partitioning may need to be considered when simulating secondary OA formation in NCP.

## 1 Introduction

Organic aerosols (OA) account for a substantial mass fraction of atmospheric fine particulate matter (Jimenez et al., 2009). However, the simulation results (e.g. concentrations and oxidation states) from chemical transport models often fail to agree with the observations to a certain degree (Chen et al., 2011;Matsui et al., 2009), which is partly due to our

limited understanding of chemical mechanisms, reaction rates, and lifetime of OA. Volatility and viscosity are important properties of OA. They have substantial impacts on the gas-particle partitioning of oxidized products (Shiraiwa and Seinfeld, 2012;Liu et al., 2018) and consequently the formation and evolution of OA, which further contributes to the uncertainty in predictions of aerosol related air quality and climate effects (Glasius and Goldstein, 2016;Shrivastava et al., 2017).

The OA volatility can be quantified by various approaches. Compared to estimations based on elemental formulas and measured partitioning of gas/particle species, thermogram analysis was found to be the most reproducible (Stark et al., 2017). As a result, the thermodenuder (TD) combined with high-resolution time-of-flight aerosol mass spectrometer (HR-AMS) has been widely used for quantification of OA volatility. Laboratory researches provide the characterization of specific secondary OA (SOA) volatility and the effect of temperature and relative humidity (RH).

For example, the evaporation kinetics of limonene SOA particles at lower RH levels (<5% and 50%) are nearly the same, while a slightly larger fraction evaporates at 90% RH (Wilson et al., 2015;Lee et al., 2011b). Zaveri et al. (2020) found that the aged α-pinene SOA had a higher volume fraction remaining (VFR) than fresh SOA under high TD temperature. Compared with laboratory experiments, the oxidation pathways and oxidants are far more complex in ambient air, and the oxygenated products can be composed of hundreds or even thousands of species with a wide range

of volatilities. Previous field observations on volatility distributions of OA are mainly focused on Europe and U.S. under low $NO_x$ levels (Xu et al., 2016;Louvaris et al., 2017a;Saha et al., 2017;Kostenidou et al., 2018). Lee et al. (2011a) found that $NO_x$ can have a large impact on the volatility of SOA in chamber experiment, suggesting that the OA volatilities in high $NO_x$ and high particulate matter (PM) environment, e.g., North China Plain (NCP), need to be further investigated. The volatility of OA also presents strong seasonal variations. For example, Huang et al. (2019)

found that OA in winter is less volatile than that in summer in Germany. The volatility of cooking OA (COA) and hydrocarbon-like OA (HOA) at the same sampling site varied substantially between summer and winter (Saha et al.,





2018;Paciga et al., 2016). These differences can be attributed to the differences in source emissions, precursors and temperature (Schervish and Donahue, 2020). To our knowledge, the volatility of OA has only been characterized in summer in NCP (Xu et al., 2019;Qiao et al., 2020), and observations in urban and rural areas during wintertime are very limited. In addition, previous studies investigated the volatility of primary OA (POA) including biomass burning, traffic and cooking emissions (May et al., 2013a;May et al., 2013b, c;Takhar et al., 2019), however, the measurements of coal combustion OA (CCOA), a dominated factor of POA in NCP (Wang et al., 2019), are rare. Failing to consider the contributions of intermediate volatility organic compounds (IVOC) and semi-VOC (SVOC) from such POA factors may lead to the underestimation of SOA concentration in models. In addition, many studies show that not all aerosol evaporated even after heating to high temperatures (i.e., 230–300℃)(Massoli et al., 2015;Xu et al., 2016). Those nonvolatile compounds can contribute to new particle formation and subsequent growth (Wehner et al., 2004;Xu et al., 2016;Massoli et al., 2015;Wang et al., 2017). Despite the increasing interest on nonvolatile particles, our understanding of this type of nonvolatile particles is incomplete, especially in highly polluted environments.

The OA volatility is intrinsically related to particle phase state that plays an important role in affecting heterogeneous reactions and the formation of cloud condensation nuclei. Particle phase states have been measured by using three-arm impactor (Liu et al., 2017;Liu et al., 2019) and polarization lidar (Tan et al., 2020) in China in recent years. The results showed that particles are generally in liquid state throughout the year in south China, while there is a transition from semisolid to liquid state as RH increases above 60% in winter in NCP. However, these studies measured the phase state of bulk aerosol that is generally dominated by hygroscopic secondary inorganic species, our knowledge of the OA phase state and viscosity remains limited. Some methods are developed to estimate OA phase state based on the molar mass, molecular atomic oxygen-to-carbon ratio (O/C) of SOA components and the number of carbon, hydrogen, and oxygen atoms (Shiraiwa et al., 2017;DeRieux et al., 2018). However, the effects of molecular structure and functional groups on glass transition temperature ($T_g$), a parameter determining a phase transition between amorphous solid and semisolid states, are not considered in those studies. Recently, the close relation between volatility and viscosity have been proved (Zhang et al., 2019;Champion et al., 2019), and parameterizations are developed to predict viscosity based on O/C and volatility at 11 global sites (Li et al., 2020). However, the simulation of the phase state and viscosity of OA in NCP during wintertime have not yet been made, impeding our understanding of the phase states of OA and its potential impacts.

In this study, a HR-AMS coupled with a TD was deployed in summer and winter at an urban site in Beijing, and a rural site in winter in NCP to investigate the differences of OA volatilities in different seasons and chemical environments. The volatility distributions of primary and secondary OA factors are estimated, and the impacts of RH are elucidated.



Further, the glass transition temperature and viscosity of OA at urban and rural sites are estimated by using the recently developed parameterization, and their implications in phase state and gas-particle partitioning are demonstrated.

## 2 Experimental methods

### 2.1 Measurements

The measurements were conducted at an urban site (Institute of Atmospheric Physics (39°58'N, 116°22'E)), from 20 May 2018 to 23 June 2018 and from 20 November to 25 December 2018, and a rural site (Gucheng in Hebei province (39°09'N, 115°44'E)) from 10 December 2019 to 13 January 2020. A detailed description of the two sampling sites is given in Xu et al. (2015) and Sun et al. (2020). Ambient particles passed through a $PM_{2.5}$ cyclone and a nafion dryer, where aerosol particles larger than 2.5 micrometer were filtered and the remaining particles were dried. After that,
aerosol particles were sampled by a HR-AMS by switching between the TD and bypass line every 15 minutes. The settings of TD heating temperature were 50, 120 (150), and 250 ℃ in summer and winter of 2018 in Beijing. In addition, the data during the temperature ramping was also included. In total, the TD data with seven temperature gradients were obtained in Beijing. Comparatively, the TD temperature was set to increase linearly in winter at Gucheng site, leading to more data points across different temperatures. The residence time (RT) of aerosol particles in
TD was 7.4 s in summer of 2018, and 10 s in winters of 2018 and 2019 due to different plug flow rate. The TD loss (90%) was calibrated using aerosolized NaCl following the methods described by Huffman et al. (2008).

### 2.2 AMS data analysis

The HR-AMS data were analyzed by PIKA (V 1.62F). The ionization efficiency (IE) and relative ionization efficiencies (RIEs) of ammonium and sulfate were calibrated following the standard protocols (Jayne et al., 2000). The
composition-dependent collection efficiency was applied for the ambient data, while a constant value (0.5) was used for the TD data (Huffman et al., 2009). All elemental ratios of OA in this study were calculated by the "Improved-Ambient (I-A)" method (Canagaratna et al., 2015) unless specified. The combined data from bypass and TD lines ($MS_{bypass+TD}$) were analyzed with positive matrix factorization (PMF) to resolve potential OA factors (Ulbrich et al., 2009). Four factors were identified in summer of 2018, including HOA, COA, less oxidized oxygenated OA
(LO-OOA) and more oxidized OOA (MO-OOA). In winter of 2018 in Beijing, fossil fuel related OA (FFOA) and oxidized POA (OPOA) were also identified in addition to the COA, LO-OOA and MO-OOA. Compared with Beijing, four OA factors including HOA, coal combustion OA (CCOA), BBOA and OOA were identified at Gucheng site in winter. It should be noted that FFOA in winter in Beijing refers to the mixed the HOA and CCOA which cannot be



separated by PMF. A detailed description of the source apportionment of OA at the two sites is given in Xu et al. (2019) and Chen et al. (in preparation).

**2.3 Estimation of OA volatility distribution**

A detailed description for estimation of the atmospheric organic aerosol volatility distribution is given in Karnezi et al. (2014). Briefly, six logarithmically spaced effective saturation concentration ($C^*$) bins with the maximum value of 100 μg m$^{-3}$ are used to fit the measured thermograms since that there are little information on the partitioning of compounds with $C^* \geq 1000$ μgm$^{-3}$ due to the average OA concentration being 13-23 μg m$^{-3}$ in this study. In addition, 6 discrete values of vaporization enthalpy and accommodation coefficient were used, i.e., 20, 50, 80, 100, 150, and 200 kJ mol$^{-1}$, and 0.01, 0.05, 0.1, 0.2, 0.5, and 1, respectively (Karnezi et al., 2014). The choice of $C^*$ bins depends on the best fits between the measured and predicted thermogram. In this study, the combinations of all properties with the smallest error (top 1%) were identified as "best estimate". The predicted and absolute thermograms are shown in Figs. S1-S2. The mass fraction of each $C^*$ bins ranged from 0 to 1 with a step of 0.1.

**2.4 Predictions of the glass transition temperature and viscosity**

A detailed description of predicting the glass transition temperature, viscosity and some parameters of OA is given in Li et al. (2020). Briefly, $T_{g,i}$ for each volatility bin is predicted based on volatility and O/C (A-A method) using Eq. (1)

$$T_{g,i} = 289.10 - 16.50 \times \log_{10}(C_i^0) - 0.29 \times \left[\log_{10}(C_i^0)\right]^2 + 3.23 \times \log_{10}(C_i^0)(O/C) \qquad (1)$$

The term $C^0$ here refers to $C^*$ based on the assumption of ideal thermodynamic mixing (Donahue et al., 2011). The glass transition temperatures of organic aerosols under dry conditions ($T_{g,org}$) is calculated by the Gordon-Taylor equation assuming the Gordon-Taylor constant ($k_{GT}$) of 1 (Dette et al., 2014).

$$T_{g,org} = \sum_i \omega_i T_{g,i} \qquad (2)$$

where $\omega_i$ is the mass fraction in particle phase for each volatility bin.

The $T_g$ of organic-water mixtures ($T_g(\omega_{org})$) at a given RH can be estimated using the Gordon-Taylor equation:



$$T_g(\omega_{org}) = \frac{(1 - \omega_{org})T_{g,w} + \frac{1}{k_{GT}}\omega_{org}T_{g,org}}{(1 - \omega_{org}) + \frac{1}{k_{GT}}\omega_{org}} \qquad (3)$$

Where $T_{g,w}$ is the glass transition temperature of pure water (136 K), and $k_{GT}$ is the Gordon–Taylor constant for organic–water mixtures which is suggested to be 2.5. $\omega_{org}$ is the mass fraction of organics in particles of organic-water mixtures, and the water content in OA can be estimated using the effective hygroscopicity parameter ($\kappa$), which is calculated by the method in Lambe et al. (2011) and Mei et al. (2013) marked as $\kappa$ (Lambe) and $\kappa$ ($f_{44}$), respectively.

Viscosity can then be estimated by applying the Vogel–Tammann–Fulcher equation $\eta = \eta_\infty e^{\frac{T_0 D}{T - T_0}}$, where $\eta_\infty$ is the viscosity at infinite temperature ($10^{-5}$ Pa s), $D$ is the fragility parameter which is assumed to be 10, and $T_0$ is the Vogel temperature calculated as $T_0 = \frac{39.17 T_g}{D + 39.17}$. When $T_g(\omega_{org})$ is larger than ambient $T$, particles are considered as solid.

The characteristic timescale of mass transport and mixing by molecular diffusion ($\tau_{mix}$) is also calculated: $\tau_{mix} = d_p^2/(4\pi^2 D_b)$, where $d_p$ is the particle diameter (assuming 200 nm here), and the bulk diffusion coefficient $D_b$ is calculated from the predicted viscosity by the fractional Stokes–Einstein relation: $D = D_C(\frac{\eta_C}{\eta})^\xi$, where $\xi$ is an empirical fit parameter, and $\xi$ =0.93. $\eta_C$ is the viscosity at which the Stokes-Einstein relation and fractional Stokes-Einstein relation predict the same diffusion coefficient.

## 3 Results and discussion

### 3.1 Volatility of aerosol species

Figure 1 shows the thermograms of NR-PM$_1$ species at both urban and rural sites. The remaining organics loading in Gucheng was lower than that in Beijing under the same TD temperature during wintertime, particularly at $T >$150 °C, suggesting that OA in Gucheng was overall more volatile than those at urban sites. Such differences can be reasonably attributed to the different OA composition at the two sites. For example, OA at the rural site presented much higher contributions from coal combustion and biomass burning emissions than urban site (Sun et al., 2020). Further, SOA composition could also be different. While photochemical aqueous-phase reactions was found to play an important role



in SOA formation in Gucheng (Kuang et al., 2020), both photochemical and aqueous-phase production were important in Beijing (Xu et al., 2017). Despite a shorter TD residence time in summer, more remaining nitrate was observed which was more likely caused by the less volatile nitrate, (e.g., organic nitrates (ON) that cannot be distinguished from inorganic nitrate with AMS). By using the $NO_X$ method (Farmer et al., 2010), we estimated that ON can account for

11-27% of total nitrate in summer, while their contributions were negligible during wintertime in both Gucheng and Beijing. Such seasonal differences in ON is in good accordance with previous observations in China (Yu et al., 2019), emphasizing the role of ON in summer. There was 40% of the residual mass of chloride in summer after heating at $T > 200$ °C, which is larger than that in winter at both urban and rural sites (4-8%). Such different residual loadings are likely due to the different sources of chloride which were mainly associated with biomass burning and coal combustion

emissions in winter, while a considerable fraction existed in the form of less volatile chloride salts (e.g., KCl) in summer.

At $T > 150$°C, sulfate in Gucheng showed lowest residual mass compared to that in Beijing, while the behaviors are contrary at $T < 150$°C. One explanation is that the different formation mechanisms (gas-phase, heterogeneous or aqueous-phase chemistry) led to the variations in mixed state which could affect the thermograms of sulfate during

three campaigns. The different contribution of organosulfates (OSs) compounds with different volatility to the total $SO_4$ is another possible reason, which is supported by the fact that the particles under different TD temperature fell into different regions in triangle-shaped space (Fig. S3) defined by Chen et al. (2019) for organic and inorganic sulfate species. All these differences in aerosol volatility between urban and rural sites emphasize the influences of non-volatile inorganic components on species measured by HR-AMS.

**3.2 Volatility of OA species**

OA was dominated by SOA in summer in Beijing (72%), and the contribution was much higher than that in winter in Beijing (42%) and Gucheng (51%), in agreement with previous studies (Zhou et al., 2020). The $C^*$ of OA in summer was 0.55 μg m$^{-3}$ in Beijing, which is smaller than that in winter in Beijing (0.71 μg m$^{-3}$) and Gucheng (0.75 μg m$^{-3}$), indicating the more volatile nature of OA in winter. One explanation was due to the higher contributions of POA in

winter which is generally more volatile than SOA (Huffman et al., 2009). This feature is contrast to that in Germany (Huang et al., 2019), where organics are found to be more volatile in summer. Such discrepancy is likely due to the different OA composition in different chemical environment. A support for this possibility is higher O/C of OA in Germany while lower O/C in Beijing during wintertime compared to summer. Enhanced primary emission sources in winter with relatively high volatility are another possible cause.



Despite HOA, CCOA, and FFOA are all related to fossil fuel, they differ in the order of volatility between urban and rural sites. HOA in Beijing showed lower saturation concentration compared to that in Gucheng ($C^*$ = 0.75 vs. 0.93 µg m$^{-3}$), while the $C^*$ of FFOA in Beijing was relatively higher compared to that of CCOA in Gucheng (1.41 vs. 0.86 µg m$^{-3}$), corresponding to lower mass fraction remaining (MFR) at the same TD temperature (Fig.2). One reason was likely due to the different quality of fuels used at urban and rural sites. Another reason could be due to the fact that FFOA in Beijing was mainly from regional transport and was aged before arriving at Beijing. FFOA and CCOA showed lower remaining loadings (~1%) compared to HOA (8-10%) at $T > 200°C$, implying that HOA contained more non-volatile compounds. Polycyclic aromatic hydrocarbons (PAHs, compounds dominantly from coal combustion) showed a contribution of SVOC by 60-71%, consistent with the high contributions of fossil sources to more volatile organic aerosol based on radiocarbon-based ($^{14}$C) approach (Ni et al., 2019). It should be noted that OA factors related to fossil fuel combustion also showed a considerable contribution of extremely low-volatility compounds (ELVOCs with $C^* \leq 10^{-4}$ µg m$^{-3}$) (5-13%), which is comparable to that in Paris (11%) (Paciga et al., 2016).

The fraction of low-volatility compounds (LVOC) in COA in winter (44%) is slightly higher than that in summer (40%) in Beijing, which fell in the range of unoxidized (54%) and ozonolysis of canola oil (29%) (Takhar et al., 2019), and comparable to that in previous field studies (Paciga et al., 2016;Louvaris et al., 2017b). Higher $C^*$ of COA in summer than that in winter (0.79 µg m$^{-3}$ vs. 0.59 µg m$^{-3}$) indicated the less volatile properties likely due to the different cooking types. For example, barbecue is popular in summer, yet not in winter due to low ambient temperature. The SVOC contributed 67% to OPOA ($C^*$ = 1.3 µg m$^{-3}$), which is in the range of $C^*$ of POA (0.6-1.4 µg m$^{-3}$) at the urban site in winter. 33% of BBOA in Gucheng evaporated at $T > 200°C$, which is comparable to BBOA in Xianghe, a rural site in NCP (Qiao et al., 2020). The contribution of SVOC (51%) in BBOA in Gucheng is lower than that measured in combustion chamber (80%) (May et al., 2013a) but overall comparable with that in Centreville, AL (47%) (Kostenidou et al., 2018). The differences in volatility distributions of BBOA was likely due to the variations in biomass fuels, combustion conditions, and the extent of atmospheric aging (Ghadikolaei et al., 2020).

Similar to previous studies (Paciga et al., 2016;Kostenidou et al., 2018), LO-OOA was more volatile than MO-OOA in summer and winter, consistent with the fact that MO-OOA dominated OA at $T > 200 °C$. SVOC accounted for 64% and 70% of LO-OOA in winter and summer, respectively with a lower $C^*$ in winter (0.78 µg m$^{-3}$ vs. 1.58 µg m$^{-3}$), highlighting that LO-OOA was more volatile in summer. Such seasonal differences can be explained by the different precursors and formation conditions of LO-OOA in two seasons, which is further supported by the differences in mass spectra. For example, the $f_{C2H3O+}/f_{CO2+}$ ratio of LO-OOA in summer was higher than that in winter. It should be noted that volatility of LO-OOA contradicted the results of thermograms which showed higher evaporation loss in winter



than that in summer (Fig.2). While the longer RT in winter is one of the causes, higher effective vaporization enthalpy (136 kJ mol$^{-1}$ vs. 157 kJ mol$^{-1}$) in winter is another reason. MO-OOA had comparable effective vaporization enthalpy (56 kJ mol$^{-1}$ vs. 58 kJ mol$^{-1}$) in summer and winter, yet showed more remaining loadings in winter at the same TD temperature with lower $C^*$ (0.49 μg m$^{-3}$ vs. 0.69 μg m$^{-3}$). Noted that LVOC compounds with $C^*$ =0.001 μg m$^{-3}$, 0.01 μg m$^{-3}$and 0.1 μg m$^{-3}$ contributed similarly to MO-OOA in summer and winter (Fig.3), indicating that the LVOC compounds of more aged SOA is independent of seasons. One reason is that long-time aging process of OA in the atmosphere could lead to similar chemical compounds in summer and winter. Compared with urban site, the remaining SOA after TD heating at the rural site fell in the range of LO-OOA and MO-OOA in Beijing, which consisted of 32% LVOC and 68% SVOC.

**3.3 Comparisons between volatile and non-volatile compounds**

We further compared aerosol composition at different TD temperatures during three campaigns. Here we defined aerosol species remaining at $T$ > 200 °C as non-volatile compounds, and those evaporated at $T$ < 90°C as volatile compounds. As shown in Fig.S4, signals for $m/z$ 100–180 (a potential indicator for oligomers) (Denkenberger et al., 2007) decreased with increasing TD temperature, suggesting that non-volatile organics are unlikely to be oligomers formed within the heated TD. The mass loadings of non-volatile sulfate and nitrate were comparable in Beijing in both summer (~0.39 μg m$^{-3}$) and winter (0.10 vs. 0.15 μg m$^{-3}$). One reason was likely due to the high contribution of low volatility organic nitrates in summer. Comparatively, the ratio of non-volatile SO$_4$ to NO$_3$ was 4.6 at the rural site during wintertime, highlighting the dominant role of sulfate in non-volatile compounds. As shown in Fig.S5, sulfate between 100 nm and 300 nm accounted for 56% of total SO$_4$ at $T$ > 200°C, which is much higher than that in ambient air (31%). One explanation is that the sulfate measured by HR-AMS has contributions from OSs, which showed a prominent peak below 320 nm (Kuang et al., 2015) with lower volatility than ammonium sulfate. Non-volatile OA accounted for 51% of the total non-volatile NR-PM$_1$ in summer, which was lower than that in winter (65-72%). This result indicated that non-volatile OA was more important in winter than summer. Note that the contribution of non-volatile OA to the total OA was comparable between summer and winter (6-8%), yet lower than that observed in Athens (Gkatzelis et al., 2016) and water-soluble non-volatile OA (Chakraborty et al., 2016) likely due to the different chemical compounds at various sites. The non-volatile OA correlated well with equivalent BC (eBC) measured by a seven-wavelength Aethalometer (AE33) ($R^2$=0.69-0.82), suggesting that it was well mixed with eBC during the aging processes in the atmosphere, consistent with the observations in Melpitz (Poulain et al., 2014) and London (Xu et al., 2016). The non-volatile OA is dominated by MO-OOA (a factor related to aqueous processes (Xu et al., 2017)), with a contribution up to 90% in winter. This result suggests that the aqueous phase processing plays an important role in





formation of non-volatile OA (e.g., diacids, and oligomers), particularly during severe haze episodes with high RH in winter in NCP (Yu et al., 2014;Ortiz-Montalvo et al., 2014;Ortiz-Montalvo et al., 2012). Such results are also supported by the large increase of non-volatile OA as elevated RH (Fig. S6), and the increases signal fraction of $CHO^+$ ($m/z$ 29) in the mass spectra (7.2%-16.1%), a tracer ion related to aqueous processes (Zhao et al., 2019).

Different from non-volatile components, the volatile compounds showed overall comparable contributions to the total volatile NR-PM$_1$ during three campaigns. Chl was an exception with higher fraction in winter (7-8%) than in summer (1%). The volatile NR-PM$_1$ was dominated by NO$_3$ (34-36%) and OA (36-41%) at both urban and rural sites, while the contribution of SO$_4$ was small (4-9%). We noticed that the composition of volatile OA was substantially different between summer and winter. As shown in Fig.4, the volatile OA was dominated by SOA (74%, mainly LO-OOA) in
summer, while POA in winter (61-62%), indicating that primary emissions played more important roles in volatile OA in winter. We noticed that FFOA was a dominant contributor of volatile POA at rural site, while COA has made an important contribution to volatile POA at urban site during wintertime.

### 3.4 Volatility of SOA under different RH levels

Figure 5 shows the thermograms of LO-OOA and MO-OOA at three different RH levels in summer and winter in
Beijing. We found that the MFR of MO-OOA as a function of TD temperature was substantially different across different RH levels. MO-OOA shows more evaporative loss at higher RH levels (RH>70%) in both summer and winter, suggesting that MO-OOA compounds formed at high RH appeared to be more volatile compared to that formed at lower RH. This result is consistent with the RH dependence of volatility for SOA in chamber experiments (Wilson et al., 2015) (Zaveri et al., 2020). Comparatively, LO-OOA showed similar changes in MFR at different RH levels,
particularly in summer, indicating that photochemical processing produced SOA with similar volatility despite the different chemical environment. Overall, our results highlight that the molecular composition of MO-OOA at different RH levels could be very different, yet their similar AMS mass spectra make it a challenge to be separated by PMF. For example, MO-OOA at low RH levels was more likely from a long-time aging in the atmosphere or aqueous-phase processing overall a regional scale that was transported to Beijing, while it could be more associated with local
aqueous-phase processing at high RH levels with stagnant meteorological conditions. A recent study by Chen et al. (2020) further supported that the SOA factors identified by AMS-PMF can be further separated into more factors with different chemical processing by using molecular compositions from chemical ionization mass spectrometer with a filter inlet for gases and aerosols (FIGAERO-CIMS) measurements. Another possibility for the RH dependence of MO-OOA volatility is that the particle phase diffusivity limited the evaporation under dry conditions (Yli-Juuti et al.,



2017;Li and Shiraiwa, 2019;Liu et al., 2016).

## 3.5 Viscosity of OA

Figure 6 shows the 2D-VBS framework of O/C vs. $\log_{10}C^*$ and the correlation between $T_{g,org}$ and $\log_{10}C^*$. The average $T_{g,org}$ of OA varied from 289.7 to 291.5 K in NCP in summer and winter, which is in the range of the values estimated by chemical composition (Slade et al., 2019;Ditto et al., 2019) and chemical transport model simulations (Shiraiwa et al., 2017) in several field campaigns. In general, $T_{g,org}$ of OA in summer in Beijing (291.5 K) is larger than that in winter (289.7-290.0 K), yet it is lower than that in Europe and the U.S. (Li et al., 2020). Such differences are caused by the fact that highly volatile OA in China facilitates the partitioning of more SVOC into the particle phase compared to megacities in Europe and U.S. (Xu et al., 2019). The $T_{g,org}$ of FFOA/CCOA, a unique OA factor in NCP, is 285.8 K in Beijing and 288.9 K in Gucheng, which is overall slightly lower compared to that of HOA (288.4-289.7 K) in China. Such differences in $T_{g,org}$ between HOA and FFOA/CCOA agree with the overall higher $C^*$ of FFOA. Even for the same OA factor, the differences in $T_{g,org}$ do exist at different sampling time and site. For example, the $T_{g,org}$ of BBOA in Gucheng (294.4 K) is lower than that in Athens, yet comparable to that in Mexico city (Li et al., 2020), which is partly attributed to the different fuels, combustion conditions and oxidation during transport leading to the differences in volatility and oxidation degree. MO-OOA showed higher $T_{g,org}$ than LO-OOA in both summer (290.2 vs. 285.5 K) and winter (292.5 vs. 289.9 K) in Beijing, consistent with the previous urban observations (e.g. Paris, Mexico City) (Li et al., 2020).

Figure 7 shows diurnal variations of predicted viscosity of OA using measured $T$ and RH during three campaigns. The predicted viscosity using different kappa values calculated by two methods correlates well with each other. The diurnal variation of viscosity is significantly affected by $T$ and RH, and thus water associated with organics. Overall, in winter of 2018 in Beijing, the OA occurs as solid with the predicted viscosity $> 10^{12}$ Pa s due to low ambient temperature and RH. The mixing time is larger than $10^3$ hours, thus kinetically limited gas-particle partitioning may need to be considered when simulating SOA formation in winter in Beijing (Li and Shiraiwa, 2019;Maclean et al., 2017;Shiraiwa et al., 2011). The viscosity of OA varies from $10^2$ to $10^6$ Pa s in Beijing in summer and from $10^3$ to $10^{11}$ Pa s in Gucheng in winter, suggesting a semi-solid phase throughout the day. The diurnal variations of predicted viscosity are characterized by increases in the afternoon in summer in Beijing and in winter in Gucheng, which is associated with diurnal variations of ambient RH and $T$. However, such diurnal variations of predicted viscosity are different from those in Amazon (Bateman et al., 2017) and Michigan (Slade et al., 2019), where enhanced viscosity at night due to the influence of biomass burning and formation of high molar mass organic compounds was observed. Note that the





viscosity of OA in Gucheng shows a large afternoon peak, while it is small in Beijing in summer. Such differences are partly caused by the differences in diurnal variations of RH that are negatively related to the rebound fraction, an indicator of the viscosity (Liu et al., 2017). For example, as shown in Fig.7, the RH shows considerable and rapid decrease from ~80% at 10:00 to 44% in the afternoon in Gucheng in winter, while the decreases in RH in Beijing during the same period of time are small (< 20%). It should be noted that we did not consider the mixing of OA and inorganic species in this work that can have influences on $T_{g,org.}$ and viscosity due to the water absorbed by inorganics (Pye et al., 2017).

**4 Conclusion**

A TD-AMS system was deployed at urban and rural sites in NCP in summer and winter to investigate the volatility and viscosity of OA. Our results showed that the $C^*$ of OA in summer in Beijing (0.55 μg m$^{-3}$) is lower than that in winter (0.71-0.75 μg m$^{-3}$), indicating that OA was more volatile in winter. One reason was due to enhanced primary emissions from coal combustion and biomass burning with high volatility. The volatility distributions of OA varied differently among different OA factors and seasons. We found that the volatility of MO-OOA was RH dependent with higher volatility at higher RH levels. These results demonstrated that the composition and formation mechanisms of MO-OOA can be significantly different under different RH levels, yet such chemical information cannot be illustrated by PMF analysis of bulk OA. Future studies combing AMS and molecular level characterization of OA can allow for deeper insights into the sources and properties of MO-OOA (Qi et al., 2019;Chen et al., 2020). We also found that the volatile properties of fossil fuel related OA were quite different between urban and rural sites, likely due to variations of oxidation during transport, different coal fuels, and combustion conditions. The compositional differences between volatile and non-volatile species were also evaluated. Our results showed that POA dominated volatile OA in winter (61%-62%), while SOA contributed more to volatile OA in summer (74%). Non-volatile OA that is dominated by MO-OOA was highly correlated with BC, and increased as a function of RH, highlighting the potential formation of aqueous-phase SOA on BC. The glass transition temperature and viscosity of OA were estimated using saturation mass concentration and atomic O/C ratio with the recently developed parameterization formula (Li et al., 2020). Our results showed that the $T_g$ of OA in summer in Beijing (291.5 K) is higher than that in winter (289.7-290.0 K), and both are overall lower than that in Europe and the U.S. The viscosity analysis suggested that OA occurred mainly as solid in winter in Beijing, and the mixing time can be as long as $10^3$ hours because of low temperature and RH, while it dominantly existed in semi-solid phase in Beijing in summer and Gucheng in winter. Our results have significant implications that kinetically limited gas-particle partitioning needs to be considered in chemical transport models when simulating SOA formation in NCP.





*Data availability.* The data in this study are available from the authors upon request (sunyele@mail.iap.ac.cn).

*Author contributions.* YS and WX designed the research. WX, CC, YQ, CX, ZL, JS, NM and WanX conducted the measurements. WX, CC, YQ, YL and ZZ analyzed the data. YL and ZZ supported the viscosity analysis, and EK and SP supported the mass transfer model analysis. YL, PF, ZW, JZ, DW and NLN reviewed and commented on the paper. WX and YS wrote the paper.

*Competing interests.* The authors declare that they have no conflict of interest.

*Acknowledgements.* This work was supported by the National Natural Science Foundation of China (41975170, 91744207), and the Beijing Municipal Natural Science Foundation (8202049).

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


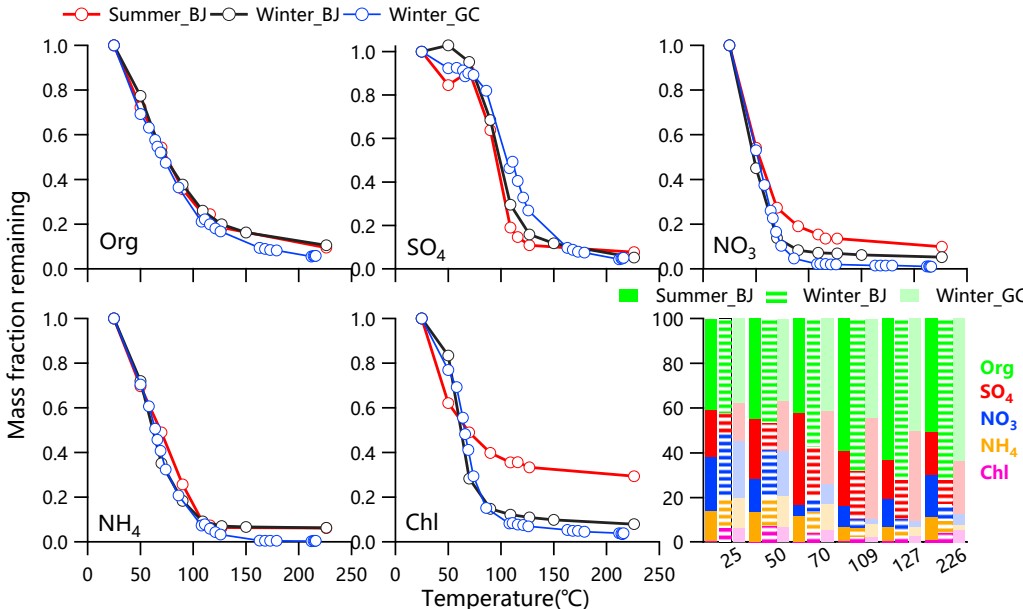

**Figure 1. Thermograms of non-refractory submicron aerosol species including organics (Org), sulfate (SO₄), nitrate (NO₃), ammonium (NH₄) and chloride (Chl). The mass fractions of size-resolved nonrefractory submicron aerosol (NR-PM₁) species as a function of TD temperature are also shown.**

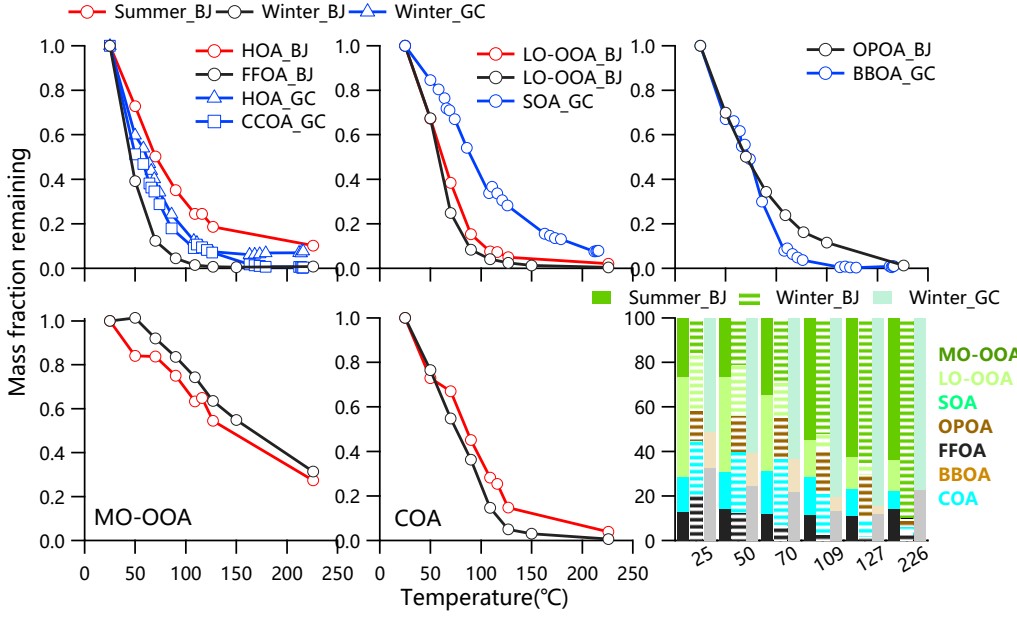

**Figure 2. Thermograms of OA factors at both urban and rural sites. The mass fractions of OA factors as a function of TD temperature are also shown.**




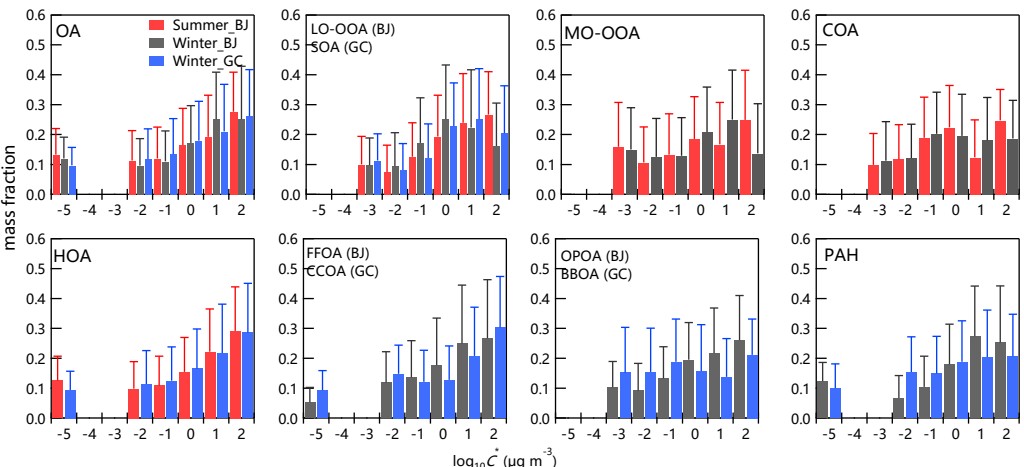

**Figure 3. Predicted volatility distributions of OA, OA factors and PAH. The error bars are the uncertainties derived using the approach of Karnezi et al. (2014)**

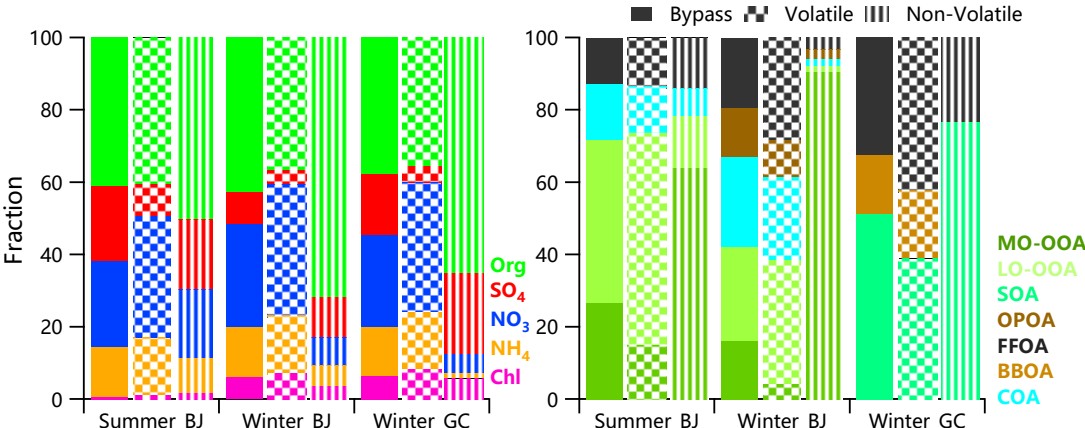

5  **Figure 4. Average composition of total, volatile, non-volatile PM and OA in Beijing and Gucheng.**



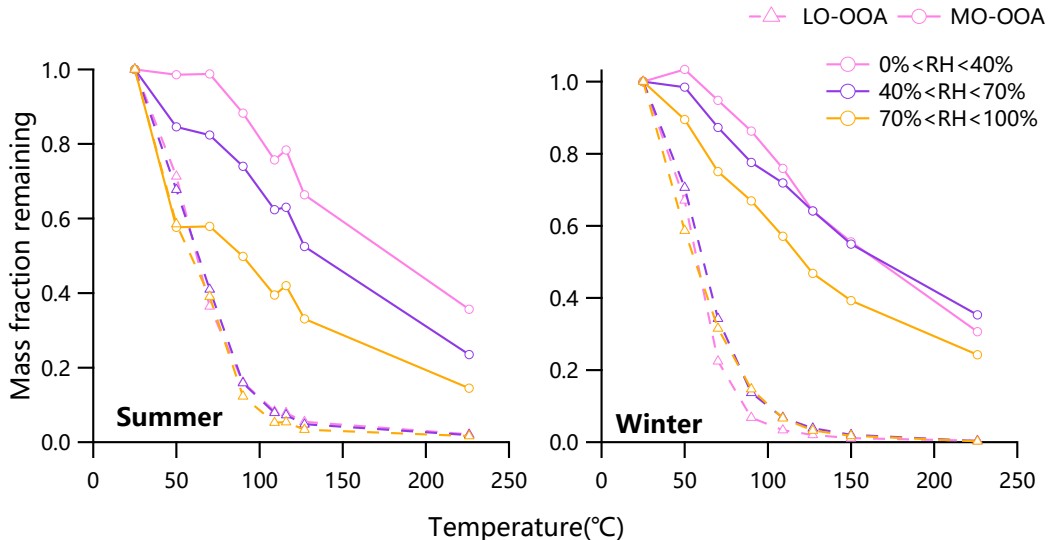

**Figure 5. Thermograms of LO-OOA and MO-OOA during different RH levels in summer and winter in Beijing.**

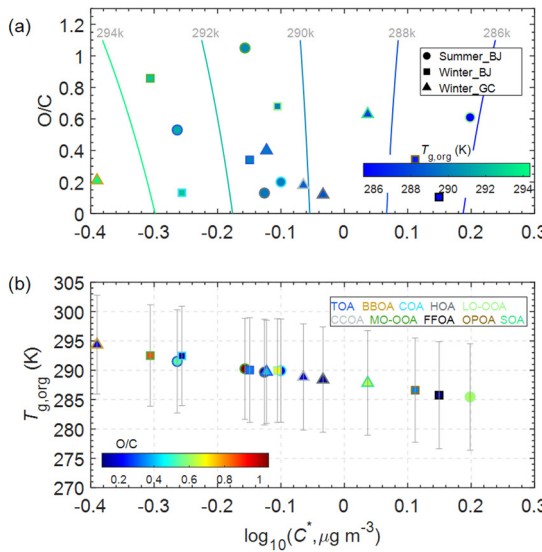

**Figure 6. Predicted glass transition temperatures of organic aerosols under dry conditions ($T_{g,org}$). The fill color of the**
5   **markers represents $T_{g,org}$ in (a) and O/C in (b). The marker edge color indicates the OA components identified by**





PMF.

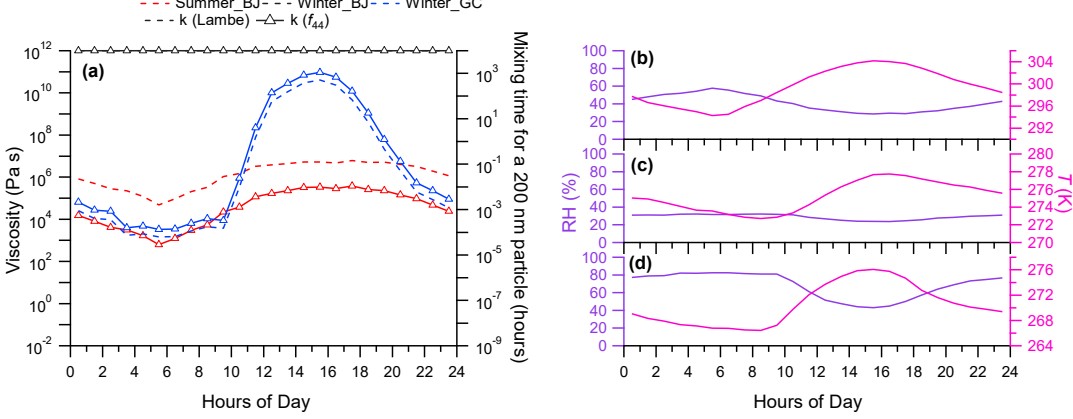

**Figure 7. Diurnal variations of (a) viscosity of total OA and ambient RH and $T$ in Beijing in (b) summer and (c) winter. The diurnal cycles of ambient RH and $T$ in Gucheng during wintertime are shown in (d). Characteristic mixing timescales of organic molecules with a radius of $10^{-10}$ m within 200nm particles are also shown on the right axis.**