# Peer review of "Organic aerosol volatility and viscosity in North China Plain"

_Atmospheric Chemistry and Physics, 2020_

## Referee Comment (RC1) · Anonymous Referee #1 · 24 Dec 2020

General Comments: The manuscript by Xu et al. investigated aerosol volatility and viscosity in North China Plain (NCP) using thermodenuder coupled with high resolution aerosol mass spectrometer. Generally, volatility distributions and glass transition temperature were compared in detail during two different seasons (summer and winter) at two different locations (urban site (Beijing) and rural site (Gucheng)). The compositional differences between volatile and non-volatile species were also elucidated. They found that OA in winter in NCP is more volatile than that in summer. In addition, OA occurred mainly as solid in winter in Beijing, while it dominantly existed in semi-solid phase in Beijing in summer and Gucheng in winter. The topic fits well within the scope of ACP. This manuscript is generally well written. Before its publication, the following comments need to be addressed.

[Figure]

Specific Comments: 1. The predicted viscosity was partly influenced by the low RH (overall lower than ~40% throughout the day in winter in Beijing). Can you also provide the predicted viscosity under high RH in winter in Beijing? For example, I noticed you have the high RH data in Fig.5. 2. Page 9, line 15: The high contribution of low volatility organic nitrates in summer is the reason of higher concentrations of non-volatile nitrates in summer, rather than the comparable non-volatile sulfate and nitrate. Please elaborate. 3. MFR (an extensive parameter explicitly depending on mass concentration) is not equivalent to volatility since volatility is an intensive parameter, which only depends on chemical nature of compounds in a mixture. However, MFR was used as a basis for volatility comparison in Page 10, line 17. In addition, are those differences in thermograms between LO-OOA during different RH levels and that of MO-OOA caused by concentration? 4. How did you measure PAH? Please describe in the measurement or data analysis section. In addition, are there any specific reasons that you used C* =10 -5 $\mu$gm−3 to fit the measured thermograms? 5. The mass fractions of PM and OA species in Figs. 1-2 : It looks like the scale of the x-axis is wrong. The ambient temperature is obviously not 25 âĎĊ in winter. 6. I suggest putting mass concentrations of composition of total, volatile, non-volatile PM and OA in a table so that the readers can see them clearly.

---

## Referee Comment (RC2) · Anonymous Referee #2 · 4 Jan 2021

This manuscript used TD-AMS system to study the volatility of organic aerosol at three sites in North China Plain. Further, the inferred volatility distribution together with literature parameterization is used to infer the aerosol viscosity and glass transition temperature. While the data analysis is solid and the discussions are thorough, I find it challenging to grasp the key messages from the manuscript. As the volatilities of several OA factors measured at two sites (rural vs urban) in two seasons (summer vs winter) are contrasted in the manuscript, the discussions are rather scattered. I appreciate the authors' efforts to present such a comprehensive dataset, but I humbly suggest the authors to organize the discussions/conclusions in a more coherent and systematic way, or better emphasize the significant findings from this study, which will better convey the crucial findings and increase the impact of this study.

[Figure]

Major Comments 1. Besides re-organizing the manuscript, another suggestion I have which may distinguish this study from other similar TD-AMS studies is regarding the RH-dependent volatility of MO-OOA, which I find to be one of the most interesting findings in this study. It is intriguing why the volatilities MO-OOA and LO-OOA exhibit such RH-dependence. Although possible reasons are discussed, the authors are biased to "chemical composition" as stated in Conclusion section (Page 12 Line 13-16). However, the RH-driven particle diffusivity is another highly possible explanation. In fact, this hypothesis can be experimentally tested by humidifying ambient aerosol before sending it through the TD. If possible, this test should be included in the manuscript. Without providing further evidence, the conclusion that the composition and formation mechanisms of MO-OOA are different under different RH is not supported. 2. The fitting of OA volatility distribution based on MFR should be elaborated. I list several questions that confuse me. (1) Are the vaporization enthalpies and accommodation coefficients values fixed or treated as tuning parameters? If the former, how sensitive are the VBS to these parameters? Also, please elaborate why these parameters derived from another study are applicable. (2) Do the authors use the MFR under different T of the whole dataset to fit a campaign-average VBS, as shown in Figure 3? (3) How is $C^*$ of OA or OA factors calculated? For example, Page 7 Line 22 mentioned that the $C^*$ of OA in summer was 0.55 ug/m3. Is this volatility-bin weighted $C^*$? Similarly, how are the effective vaporization enthalpies of OA factors calculated (Page 9 Line 2)? 3. Page 8 Line 2,3, etc. 0.75 vs 0.93 ug/m3. Please provide the uncertainty range of the estimated $C^*$ to justify if the comparison is significant or not.

Minor Comments 1. Page 12 Line 16. Typo. Replace "combing" with "combining".

---

## Author Comment (AC1) · 6 Feb 2021

We are thankful to the two referees for their thoughtful and constructive comments which help improve the manuscript substantially. Following the reviewers' suggestions, we have revised the manuscript accordingly. Listed below are our point-by-point responses in blue to each comment that is repeated in italic.

**Response to Reviewer #1**

*General Comments:*
*The manuscript by Xu et al. investigated aerosol volatility and viscosity in North China Plain (NCP) using thermodenuder coupled with high resolution aerosol mass spectrometer. Generally, volatility distributions and glass transition temperature were compared in detail during two different seasons (summer and winter) at two different locations (urban site (Beijing) and rural site (Gucheng)). The compositional differences between volatile and non-volatile species were also elucidated. They found that OA in winter in NCP is more volatile than that in summer. In addition, OA occurred mainly as solid in winter in Beijing, while it dominantly existed in semi-solid phase in Beijing in summer and Gucheng in winter. The topic fits well within the scope of ACP. This manuscript is generally well written. Before its publication, the following comments need to be addressed.*
We thank the reviewer's comments and have revised the manuscript accordingly.

*Specific Comments:*
*1. The predicted viscosity was partly influenced by the low RH (overall lower than ~40% throughout the day in winter in Beijing). Can you also provide the predicted viscosity under high RH in winter in Beijing? For example, I noticed you have the high RH data in Fig.5.*

[Figure]

Figure R1. Predicted viscosity of total OA measured in winter in Beijing as a function

of RH.

It is a good point. In the revised manuscript, we estimated the viscosity of total OA as a function of RH in winter in Beijing by using the average $T_g$, O/C, mass loading of OA, $T$ and density of OA in this study. The results showed that the viscosity of OA varied from $10^2$ Pa s to $10^{12}$ Pa s when RH was in the range of 40 - 85%, and it was less than $10^2$ Pa s at RH>~85%, suggesting that OA particles in winter in Beijing would exist mainly as semi-solid phase and mostly liquid at RH = 40% - 85% and RH>85%, respectively.

Following the reviewer's comments, we expanded the discussions on viscosity at high RH.
"We further explored the viscosity of OA as a function of RH in winter in Beijing and found that the viscosity of OA varied from $10^2$ Pa s to $10^{12}$ Pa s as RH was in the range of 40 – 85%, and it was less than $10^2$ Pa s at RH>85%. These results suggest that OA particles in winter in Beijing would exist mainly as semi-solid phase and mostly liquid at RH = 40% - 85% and RH>85%, respectively."

*2. Page 9, line 15: The high contribution of low volatility organic nitrates in summer is the reason of higher concentrations of non-volatile nitrates in summer, rather than the comparable non-volatile sulfate and nitrate. Please elaborate.*

We deleted this sentence following the reviewer's suggestion.

*3. MFR (an extensive parameter explicitly depending on mass concentration) is not equivalent to volatility since volatility is an intensive parameter, which only depends on chemical nature of compounds in a mixture. However, MFR was used as a basis for volatility comparison in Page 10, line 17. In addition, are those differences in thermograms between LO-OOA during different RH levels and that of MO-OOA caused by concentration?*
We agree with the reviewer that the volatility comparison should not be made based on MFR.
We have revised the corresponding context, as follows:
", suggesting that MO-OOA compounds formed at high RH contained more relatively high volatility compounds compared to that formed at lower RH."

Table R1. A summary of average mass concentrations ($\mu$g m$^{-3}$) of MO-OOA and LO-OOA at different RH levels.

|  | LO-OOA$_{Summer}$ | MO-OOA$_{Summer}$ | LO-OOA$_{Winter}$ | MO-OOA$_{Winter}$ |
|---|---|---|---|---|
| RH<40% | 4.3 | 1.9 | 2.4 | 1.4 |
| 40%<RH<70% | 7.4 | 3.9 | 7.9 | 5.5 |
| RH>70% | 6.2 | 5.7 | 16.7 | 9.3 |

Table R1 shows the mass concentrations of LO-OOA and MO-OOA at different RH

levels. Both LO-OOA and MO-OOA showed the increasing trends as a function of RH. The mass loadings of LO-OOA at high RH levels were ~6.5 times higher than that at low RH levels in winter in Beijing. Such large increases were comparable with those in MO-OOA in winter in Beijing. Comparatively, the thermograms between LO-OOA and MO-OOA show significant differences at different RH levels. These results imply that the differences in thermograms between LO-OOA and MO-OOA were unlikely caused by the mass concentrations.

*4.How did you measure PAH? Please describe in the measurement or data analysis section. In addition, are there any specific reasons that you used C\*=10 -5 µgmm3 to fit the measured thermograms?*

PAHs were determined using the approach developed by Dzepina et al. (2007). PAHs are very resistant to fragmentation after ionization, and therefore are often the largest peaks in mass spectra above *m/z* 200. Dzepina et al. (2007) then developed a method to subtract the non-PAH signals from the *m/z*'s of PAHs. The sum of PAH-related *m/z*'s is converted to mass concentrations after considering collection efficacy and ionization efficiency. The detailed method for quantification of PAHs is given in Dzepina et al. (2007).

Following the reviewer's comments, we added

", which were determined with the approach recommended by Dzepina et al. (2007)"

[Figure]

Figure R2. Thermograms of PAH measured by TD-AMS in winter of 2018 in Beijing using six logarithmically spaced $C^*$ bins including 100, 10, 1, 0.1, 0.01 and (a) 0.001 µg m$^{-3}$ (or (b) 0.0001 µg m$^{-3}$ and (c) 0.00001 µg m$^{-3}$). The solid circles represent the measurements and the error bars are one standard deviation. The black lines refer to the best-predicted MFR using the algorithm of Karnezi et al. (2014).

Different volatility ranges were chosen for PAHs based on the best fits between the measured and predicted thermograms. As shown in Fig. R2, the predicted and measured MFR at the highest TD temperature showed significant differences in using minimum $C^*$ bins with 0.001 and 0.0001 µg m$^{-3}$, while the predicted MFR using minimum $C^*$ bins with 0.00001 µg m$^{-3}$ can capture better the ending points of the thermograms. Hence, 0.00001 µg m$^{-3}$ was chosen as minimum $C^*$ bins.

*5.The mass fractions of PM and OA species in Figs. 1-2 : It looks like the scale of the*

*x-axis is wrong. The ambient temperature is obviously not 25 â ˇD ˇC in winter.*
Revised.

*6.I suggest putting mass concentrations of composition of total, volatile, non-volatile PM and OA in a table so that the readers can see them clearly*
Following the reviewer's comments, we added mass concentrations of total, volatile, non-volatile PM and OA species in Table S1.

**Response to Reviewer #2**

*This manuscript used TD-AMS system to study the volatility of organic aerosol at three sites in North China Plain. Further, the inferred volatility distribution together with literature parameterization is used to infer the aerosol viscosity and glass transition temperature. While the data analysis is solid and the discussions are thorough, I find it challenging to grasp the key messages from the manuscript. As the volatilities of several OA factors measured at two sites (rural vs urban) in two seasons (summer vs winter) are contrasted in the manuscript, the discussions are rather scattered. I appreciate the authors' efforts to present such a comprehensive dataset, but I humbly suggest the authors to organize the discussions/conclusions in a more coherent and systematic way, or better emphasize the significant findings from this study, which will better convey the crucial findings and increase the impact of this study.*

We thank the reviewer's comments and have revised the manuscript accordingly. We have re-organized the conclusions according to the sequences of discussions.

*Major Comments*

*1. Besides re-organizing the manuscript, another suggestion I have which may distinguish this study from other similar TD-AMS studies is regarding the RH-dependent volatility of MO-OOA, which I find to be one of the most interesting findings in this study. It is intriguing why the volatilities MO-OOA and LO-OOA exhibit such RH-dependence. Although possible reasons are discussed, the authors are biased to "chemical composition" as stated in Conclusion section (Page 12 Line 13-16). However, the RH-driven particle diffusivity is another highly possible explanation. In fact, this hypothesis can be experimentally tested by humidifying ambient aerosol before sending it through the TD. If possible, this test should be included in the manuscript. Without providing further evidence, the conclusion that the composition and formation mechanisms of MO-OOA are different under different RH is not supported.*

We agree with the reviewer that RH-driven particle diffusivity is another possible explanation. Hence, we stated in Page10, Line 29: "Another possibility for the RH dependence of MO-OOA volatility is that the particle phase diffusivity limited the evaporation under dry conditions."

Concerning the test by humidifying ambient aerosol, we did not try it. The major reason is that aerosol particles need to be dried before sampling into TD and AMS because 1) relative humidity affects the collection efficiency of aerosol particles in AMS, and 2) the orifice is easily clogged at high relative humidity. We totally agree with the reviewer that the conclusion needs more evidence. In the future, more advanced instruments, e.g., FIGAERO-CIMS or EESI-CIMS that provide real-time measurements of molecular compositions can be deployed simultaneously to investigate the changes of OA composition as a function of relative humidity.

Previous chamber experiments have explored the particle diffusivity under different RH. For example, Zaveri et al. (2020) found that the aged α-pinene SOA (after ~16 h of photochemical aging) showed similar evaporation rate under both dry (RH < 5%) and humid (RH = 75%) conditions. Wilson et al. (2015) found that the fresh limonene SOA showed comparable volume fraction remaining (VFR) under all conditions. In addition, SOA particles contain significant amounts of oligomers (Kalberer et al., 2004;Reynolds et al., 2006;Yasmeen et al., 2010), which can retard diffusion and slow evaporation of small molecules (Widmann et al., 1998). These results together suggest that the differences in RH-dependent volatility of aged SOA were not likely caused by the RH-driven particle diffusivity.

*2.The fitting of OA volatility distribution based on MFR should be elaborated. I list several questions that confuse me. (1) Are the vaporization enthalpies and accommodation coefficients values fixed or treated as tuning parameters? If the former, how sensitive are the VBS to these parameters? Also, please elaborate why these parameters derived from another study are applicable. (2) Do the authors use the MFR under different T of the whole dataset to fit a campaign-average VBS, as shown in Figure 3? (3) How is C\* of OA or OA factors calculated? For example, Page 7 Line 22 mentioned that the C\* of OA in summer was 0.55 ug/m3. Is this volatility-bin weighted C\*? Similarly, how are the effective vaporization enthalpies of OA factors calculated (Page 9 Line 2)?*

The vaporization enthalpies and accommodation coefficients values from another study are not applicable in this study. Indeed, the estimated OA volatility was sensitive to the assumed values of effective vaporization enthalpy and mass accommodation coefficient (Riipinen et al., 2010). Hence, the vaporization enthalpies and accommodation coefficients were not fixed in this study. As we mentioned in section 2.3, the 6 discrete values of vaporization enthalpy (20, 50, 80, 100, 150, and 200 kJ mol$^{-1}$) and accommodation coefficient (0.01, 0.05, 0.1, 0.2, 0.5, and 1) were used to fit the measured thermograms. Note that Karnezi et al. (2014) tried other discretization for the values of the vaporization enthalpy and the mass accommodation coefficient, and they found minor effects on the results. As a result, we derived 96516 different results by fitting the TD data in each campaign. The combinations of all properties with the smallest error (top 1%) were chosen to calculate the "best estimate" following the methods described in Karnezi et al. (2014).

Yes, we used the MFR under different $T$ of the whole dataset to fit a campaign-average VBS.

$C^*$ was calculated by

$$log_{10}(C*) = \sum_i f_i log_{10}(C*)$$

$f_i$ is the fraction in each $C^*$ bins in above "best estimate" (top 1%).

The effective vaporization enthalpies of OA factors were calculated by

$$\Delta H = \frac{\sum_j^N [(\Delta H_j)(\frac{1}{E_j})]}{\sum_j^N [\frac{1}{E_j}]}$$

$\Delta H_j$ is the vaporization enthalpy of OA factor for data point $j$ in above "best estimate" (top 1%). $E_j$ is the percentage error.

A detailed description of the estimation of atmospheric organic aerosol volatility distribution is given in Karnezi et al. (2014) that was cited in the manuscript.

   *3. Page 8 Line 2,3, etc. 0.75 vs 0.93 ug/m3. Please provide the uncertainty range of the estimated C\* to justify if the comparison is significant or not.*
Revised.

*Minor Comments 1. Page 12 Line 16. Typo. Replace "combing" with "combining".*
Corrected.

References

Dzepina, K., Arey, J., Marr, L. C., Worsnop, D. R., Salcedo, D., Zhang, Q., Onasch, T. B., Molina, L. T., Molina, M. J., and Jimenez, J. L.: Detection of particle-phase polycyclic aromatic hydrocarbons in Mexico City using an aerosol mass spectrometer, Int. J. Mass Spectrom., 263, 152-170, 2007.

Kalberer, M., Paulsen, D., Sax, M., Steinbacher, M., Dommen, J., Prevot, A. S. H., Fisseha, R., Weingartner, E., Frankevich, V, Zenobi, R., and Baltensperger, U.: Identification of polymers as major components of atmospheric organic aerosols, Science, 303, 1659-1662, 2004.

Karnezi, E., Riipinen, I., and Pandis, S. N.: Measuring the atmospheric organic aerosol volatility distribution: a theoretical analysis, Atmospheric Measurement Techniques, 7, 2953-2965, 10.5194/amt-7-2953-2014, 2014.

Reynolds, J. C., Last, D. J., McGillen, M., Nijs, A., Horn, A. B., Percival, C., Carpenter, L. J., and Lewis, A. C.: Structural analysis of oligomeric molecules formed from the reaction products of oleic acid ozonolysis, Environ. Sci. Technol., 40, 6674-6681, 10.1021/es060942p, 2006.

Riipinen, I., Pierce, J. R., Donahue, N. M., and Pandis, S. N.: Equilibration time scales of organic aerosol inside thermodenuders: Evaporation kinetics versus thermodynamics, Atmos. Environ., 44, 597-607, 10.1016/j.atmosenv.2009.11.022, 2010.

Widmann, J. F., Heusmann, C. M., and Davis, E. J.: The effect of a polymeric additive on the evaporation of organic aerocolloidal droplets, Colloid Polym. Sci., 276, 197-205, 10.1007/s003960050229, 1998.

Wilson, J., Imre, D., Beranek, J., Shrivastava, M., and Zelenyuk, A.: Evaporation Kinetics of Laboratory-Generated Secondary Organic Aerosols at Elevated

Relative Humidity, Environ. Sci. Technol., 49, 243-249, 10.1021/es505331d, 2015.

Yasmeen, F., Vermeylen, R., Szmigielski, R., Iinuma, Y., Böge, O., Herrmann, H., Maenhaut, W., and Claeys, M.: Terpenylic acid and related compounds: precursors for dimers in secondary organic aerosol from the ozonolysis of α- and β-pinene, Atmos. Chem. Phys., 10, 9383-9392, 10.5194/acp-10-9383-2010, 2010.

Zaveri, R. A., Shilling, J. E., Zelenyuk, A., Zawadowicz, M. A., Suski, K., China, S., Bell, D. M., Veghte, D., and Laskin, A.: Particle-Phase Diffusion Modulates Partitioning of Semivolatile Organic Compounds to Aged Secondary Organic Aerosol, Environ. Sci. Technol., 54, 2595-2605, 10.1021/acs.est.9b05514, 2020.